# Consumers Associate High-Quality (Fine) Wines with Complexity, Persistence, and Unpleasant Emotional Responses

**DOI:** 10.3390/foods9040452

**Published:** 2020-04-08

**Authors:** Maria Souza-Coutinho, Renato Brasil, Clarisse Souza, Paulo Sousa, Manuel Malfeito-Ferreira

**Affiliations:** 1Linking Landscape Environment Agriculture and Food (LEAF) Research Center, Instituto Superior de Agronomia, Universidade de Lisboa, Tapada da Ajuda, 1349-017 Lisboa, Portugal; maria_sc@hotmail.com (M.S.-C.); mmalfeito@isa.ulisboa.pt (M.M.-F.); 2Culture and Art Institute, Federal University of Ceara, 60356-000 Fortaleza, Brazil; rnato_br@yahoo.com.br (R.B.); clarissemsouza@gmail.com (C.S.)

**Keywords:** wine tasting, emotions, wine styles, complexity, persistence, unpleasantness

## Abstract

The conventional method for the sensory evaluation of wine is based on visual, olfactory and gustatory perceptions described by a domain-specific language. This is a complex task, requiring extensive training, which is not feasible from a consumer perspective. The objective of this study was to apply a wine tasting sheet, including sensory and emotional responses, to simplify the recognition of fine white wines by consumers. First, a panel of 15 semi-trained judges evaluated eight sensory attributes through Optimized Descriptive Profile (ODP) methodology. Then, a group of 104 consumers evaluated five white wines with different sensory characteristics using an improved emotional wine tasting sheet. The emotions and sensations most frequently associated with white wines were obtained through the Check-All-That-Apply (CATA) approach. The eight sensory attributes were significant (*p*-value < 0.05) in the distinction of wines by the ODP. Likewise, the distinction of the wines also provided significant differences in all the emotional and sensory attributes (*p*-value < 0.05). The different wine styles could be distinguished by Principal Component Analysis (PCA) using the semi-trained judges or the consumer responses. The highest score in the “global evaluation” was given to two young, fruity wines characterized by high aromatic “initial impression”. The two fine wines, including a 2004 Burgundy Pouilly-Fuissé, were the lowest rated in “initial impression” and “global evaluation”, although they were considered by the consumers among the most complex and persistent. These wines were also most frequently associated with unpleasant emotions by the CATA test. The recognition of these fine wine attributes and their incongruity with emotional responses can be used in a rapid way by professionals to explain the different wine styles to consumers.

## 1. Introduction

The recognition of wine quality by consumers is a goal shared by winemakers that seek to obtain approval for their products. This task is not easy in a complex beverage where preference is driven not only by sensory factors but also by demographic, cultural, or psychological aspects [1]. The definition of quality is also not consensus based; in the simplest way, it is regarded as the sensory properties that a wine should have to please consumers [2], which do not always seem to fit the ideal properties of the high-quality product termed “fine wine” by professionals [3].

Several authors have described the intrinsic sensory attributes associated by experts with high-quality wines, including “high flavor intensity” [4,5], “red fruit” flavors [6], or “oak” and “no astringency” [7]. By relating wine descriptive analysis performed by trained tasters with consumer preferences, it is possible to understand which sensory profiles are preferred. Overall, higher likeability scores are given to wines dominated by fresh intense flavors (fruity or flowery) and sweet/smooth mouthfeel, even among very different cultural backgrounds, countries of origin, or levels of wine consumption [8]. Several preference nuances may be added by oak flavor perception, but the overall rule is consistent with the so-called international palate.

Besides intrinsic flavor diversity, high quality refers to other perceptions that are difficult to define like harmony, balance, persistence, and complexity [9,10]. These properties are known as holistic, emergent, or synthetic properties, because they assess the wine as a whole and cannot be fully inferred from wine intrinsic chemical composition or sensory description [3]. Their evaluation seems to be difficult to understand by consumers [11], who tend to associate the concept of quality with likeability or preference [12]. This fact hampers the recognition of fine wines characterized by weak and non-fruity flavors, together with an astringent and sour mouthfeel, which are not appreciated by inexperienced tasters. Thus, it is understandable that wine tasting may be considered a hard task only achieved after years of training and familiarity with superior quality wines [13]. Thus, the difficulty of making fine wines appreciated by consumers has two sides: learning how to recognize the emergent properties and learning how to go beyond flavor unpleasantness.

Recently, affective approaches using emotional responses have been proven to be an efficient way to understand consumer behavior [14]. Keeping in mind that sense of smell contains a pleasantness evaluation [15], it is understandable that “sweetish” fruity and flowery flavors elicit positive emotions, whereas vegetal flavors, sourness, aggressive mouthfeel, and aged character are regarded as unpleasant [16,17]. Thus, fine wines, besides being difficult to describe sensorially, are also unattractive, hindering their appreciation. To reduce these limitations, Loureiro et al. [18] proposed a tasting methodology composed of emotional and sensory attributes that enables consumers to increase the likeability of classical European wines, such as the Burgundy Chablis and Pinot Noir, after a few (2–3) tasting sessions. This approach was later improved by Coste et al. [19], using red wines showing that consumers associated fine wines with reactions of unpleasantness driven by the odor, which turned into positive surprise after mouth evaluation. According to these authors, such fast learning could be explained by the contradiction between odor and in-mouth perceptions under the framework of a psychological effect described as “cognitive dissonance” [20].

The present study is an extension of Coste et al. [19]’s methodology to white wines. Therefore, the objective was to apply a tasting sheet composed by emotional and sensory descriptions in the distinction of white wines characterized by styles varying from international commercial taste to high-quality fine wines with aged character. Hopefully, consumer responses will allow us to understand consumer preferences and will provide clues on how to make the quality of these wines more easily acknowledged.

## 2. Materials and Methods

### 2.1. Focus Group Sessions

The focus group sessions followed the methodology described in the work of Coste et al. [19]. Briefly, two sessions comprising a total of 21 volunteers were held in a classroom with the tables organized in U shape. Both sessions lasted for an average of 60 min. Each session was coordinated by one moderator and three assistants. The main goal was the analysis of the emotional tasting sheet published by Loureiro et al. [18]. Four wine samples were chosen for their distinctive sensorial characteristics: (W1) Charneca de Pegões, white Moscatel Graúdo (synonym Muscat of Alexandria) 2014 from the Portuguese wine region Península de Setúbal; (W2) Roux Pére & Fils, white Chardonnay 2005, from the French wine region Pouilly-Fuissé of Burgundy; (W3) Dona Ermelinda red wine blend Reserva 2012, from the Portuguese wine region Península de Setúbal; and (W4) Roux Pére & Fils, red Pinot Noir Les Grands Epenots Premier Cru 2004 from the French wine region Pommard of Burgundy. The focus group discussions yielded an improved tasting sheet that was here used, as described by Coste et al. [19]. Appendix A shows the final material that was produced after the completion and analysis of the focus group discussions and that was further used in the consumer panel.

### 2.2. Optimized Descriptive Profile (ODP)

The Optimized Descriptive Profile (ODP) was developed by selected participants based on their ability to detect the basic tastes in water and in wine. A total of 15 judges were selected and trained for the ODP, where the wine samples corresponded to examples of the extremes of each sensory descriptor. The full procedure has already been described by Coste et al. [19].

### 2.3. Consumer Tasting

The group of consumers (*n* = 104) consisted of 67 males and 36 females, ranging from 18 to 66 years old with a mean age of 26 years, and 73% consumed wine daily or weekly. The tastings were performed in a 100 m^2^ classroom with dark benches where glasses were distributed on individual white paper mats, with natural light, room temperature (22–24 °C), and without conditioned air. The room opened at 11.00 a.m. and closed at 4.00 p.m. The bottles were refrigerated in order to be served at 16 °C. After a short introduction about the study tasks, the consumers gave oral consent. All participants were given a short description of the main attributes to be evaluated. The tasting method was the same as the one used for the ODP. The evaluation was done on paper ballots with the emotional tasting sheet and the Check-All-That-Apply (CATA) list updated after the focus group discussions. The tasting sheet was handed out along with the list of attributes resulting from the inputs of the focus groups. On these lists, participants marked all the attributes they thought correlated with the wines presented. This test was aimed at confirming the enhanced attributes, resulting from the focus groups by analyzing whether participants opted to use them or not.

The two white wines from the ODP tests were also used in this tasting. The other four white wines were selected to encompass a broad variety of sensory qualities (Table 1). W1 was a Muscat of Alexandria varietal, with intense flavor dominated by flowery terpenic notes and smooth mouthfeel, representing the international commercial warm climate style of young white wines. In the opposite style, two fine wines with aged character were chosen (W2 and W5). W2 was an aged wine recognizable by the yellow straw color, developed flavor, and medium to high acidity, produced in Beira Interior, a cooler region from central–east Portugal. W5 was chosen as an example of a cool climate wine from a reputed appellation (Burgundy), with initial reduced character. W3 was an exemplar of popular white wines from Vinho Verde appellation (the Atlantic northwest of Portugal), slightly carbonated and with a sweet finish due to containing 11 g/L residual sugar. From the same region, but a different style, W4 was a Loureiro varietal with a noticeable fruity/flowery aroma and balanced acidity, representing typical modern Portuguese white wines. The presentation order was a balanced incomplete block design with a carry-over control, using the algorithm of Hedderley and Wakeling [21].

### 2.4. Check-All-That-Apply (CATA) and Ideal Sample Testing

The 103 consumers were asked to complete a Check-All-That-Apply (CATA) questionnaire after filling in the tasting sheet. The CATA test had 25 emotional descriptive terms selected according to preliminary tests made with focus groups and published by Coste et al. (2018) [19]. The panelists were asked to mark with an “x” the CATA test terms that best described the features for each sample. The frequency of use of each term was determined by counting the number of panelists who used the same term, according to the methodology of Vidal et al. [22].

The CATA test terms were used to identify the emotional attributes of the ideal white wine, as proposed by Van Trijp et al. (2007) [23] in the ideal profile method. Consumers were required to characterize the ideal white wine using a form with the same attributes that were generated for the CATA method. At this time, the panelists answered what characteristics identified an ideal white wine, indicating the attributes that they considered appropriate.

### 2.5. Development of the Wine Taster Emotion Wheel

The wine taster’s emotion wheel was developed based on the wine mouthfeel wheel [24,25]. The emotional terms referring to wine consumption were displayed in a systematic organization, displaying the emotional profile of a sample possibly serving as a parametric device for applying tests. The lexicon proposed by Coste et al. [19] and others obtained by the focus groups were clustered in new hierarchical subcategories inside the emotional categories proposed by Laros and Steemkamp [26], according to their first expressed emotion. The terms were clustered according to the correlation value. The strong positive correlation between several terms indicated the same basic emotional subordination and, within a specific term, indicated subordination to that term. The negative correlation among terms indicated opposite affective classification. These values of the classification parameters considered both the coherence between the semantic and cultural meanings of each term and the previous hierarchical classification proposed by Laros and Steemkamp [26]. This way, the wheel was divided into positive affects (*n* = 12), negative effects (*n* = 6), and neutral effects (*n* = 3).

The emotions and their subordinated term dispositions were based as follows: the X-axis represents the variation of affect from negative values to positive values, where 0 (zero) of this axis divides both values. The Y-axis corresponds to the arousal manifestation of the autonomic nervous system, whose graduation begins at 0 (zero) of this axis, so its variance only occurs positively. The disposition of both X- and Y-axes was not scaled, serving only as a guide of emotional manifestation for wine consumption, thereby enabling an evaluation of the emotional profile of this specific sample at the specific conditions that occurred in the analysis through overlapping a radar graphic with the frequency of emotional terms of a specific sample.

Segments were color coded to facilitate ease of use by grouping attributes together that belong to the same oral quality/dimension or that are elicited by the same wine style. Attributes that were predominantly elicited by sparkling wines were colored salmon, and taste qualities were coded yellow. Non-taste sensations were colored deep green, while those that pertain to consistency attributes were coded light blue. These attributes (chewable, warm, and overwhelming), were classified as neutral emotional terms mainly by their meaning directly related to the physical perception inside the oral cavity. Integrated qualities, other than those elicited by sparkling wines, were colored purple and subordinated to negative affect.

### 2.6. Statistical Analysis

To test the hypothesis of normality, previously the Kolmogorov–Smirnov test was applied. Analysis of Variance (ANOVA) and Principal Component Analysis (PCA) were used to analyze the results of the emotional tasting. In both the ODP and the consumer panel, Tukey’s Honest Significant Difference (HSD) test was applied to all pairwise differences between means in order to detect significant differences between wine pairs. The data from the CATA were analyzed with Cochran’s Q Test to compare each combination of wine and attribute. Correspondence analysis was run to detect possible differences between the wines in their emotional profiles. Correlations between attributes were calculated through Principal Component Analysis to indicate possible relationships between attributes.

All analyses were performed with the software XLSTAT^©^ (Addinsoft, 2020; 2019.2.1 Version, Paris, France) [27]. A *p*-value of 0.05 was considered for each statistical test unless stated otherwise.

## 3. Results

### 3.1. ODP

In order to evaluate taster performance, the values of the interaction F (sample × taster) are listed in Table 2. According to Stone and Sidel [28], the interactions between samples and tasters may occur simply because some tasters use different parts of the scale to rate the intensity of an attribute and do not necessarily reflect training failure. In this work, significant interactions for all attributes were found, indicating that there were no tasters scoring the wines contrarily to the whole panel.

The wine sensory characterization by ODP is presented in Table 3. All the attributes were significantly different (*p* < 0.05) at least for each pair of wines. The differences were significant between the highest and lowest scores for a descriptor with the gradual evolution of the intermediate scores showing that wines covered a wide continuum of sensations.

The higher smell intensity and lower complexity of W1, a young fruity/flowery Muscat, is consistent with the selected style. W2 and W5 showed similar intensity but higher complexity, as expected from the aged Burgundy and Beira Interior white wines. W4 showed lower smell intensity and higher complexity when compared to the W1 Muscat, consistent with a less fruity varietal (Loureiro) from a cooler region (Vinho Verde). Another from this region was W3, a popular gasified white wine with low smell intensity and complexity. 

The in-mouth sensation revealed that the thermal perception was higher in the aged W2 and W5 wines. This Burgundy showed the higher body and astringency scores, similar to W2 and W4, whereas W1 and W3 showed lower ratings. The persistency decreased from W5 to W1, with intermediate values for the other wines, consistent with the warmer climate origin of W1.

The attributes related to flavor changes during the course of the tasting revealed that W2 was the wine that changed and persisted (smell) the most together with W5, according to the aging character of these wines. The low values for flavor changes in W1 and flavor duration in W3 were also according to these wine styles.

The PCA revealed the similarities among the wines, with W2 and W5 placed close together and W1 and W3 in the opposite quadrant of the plot (Figure 1). This clear distinction resulted from the positive score characteristics of fine wines, such as complexity, persistence, flavor changes, and flavor permanence. W4 showed intermediate characteristics, being placed in the center of the plan, corresponding to sensory features closer to the fine wines, W2 and W5. The distinction between W1 and W3 was due to the higher smell intensity of the former and the higher body and persistency of the latter, probably due to its residual sugar (11 g/L).

Overall, the ODP results showed that the semi-trained panel was able to characterize the wines in relation to perceptions that may be used to describe fine wines. Moreover, the wines showed a gradual distinction among the several descriptors, which will be useful to study the elicited emotional responses.

The Pearson correlation coefficients relating the attributes determined by the ODP (Appendix A) evidence the relationships among the different sensory perceptions. Strong correlations were obtained between (i) “smell complexity” and “evolution of the wine in the glass”, (ii) among “thermal perception” and “smell Complexity”, “astringency”, or “duration of wine fragrance”, and (iii) between “body” and “persistency”. Interestingly, high scores for all these attributes were mostly perceived in W2 and W5, corresponding with the fine wines with aged character used in this study. Accordingly, high “smell intensity” was not correlated with all the other descriptors, being a feature of the intense fruity/flowery wines, W1 and W4.

In conclusion, the ODP approach used to train the tasters, where wine references were used as scale extremes, enabled us to characterize the sensory attributes of the 5 wines, evidencing their different styles. In particular, an opposition between the popular commercial styles of W1 and W3 and the cool climate aged wines, W2 and W5, was clearly observed. W4 showed intermediate characteristics differing from W1 by having higher scores for smell complexity, body, and persistence. Further experiments were run to check if the emotional approach used by the untrained subjects could yield similar outputs in the distinction of wine styles.

### 3.2. Consumer Tasting

The scores for all the attributes of the emotional tasting sheet given by the consumers are listed in Table 4. The highest “global evaluation” scores were given to W1 and W4, which could be explained by their international commercial style, with higher scores for “initial impression” eliciting higher “expectation for the mouth” and “overall taste evaluation” scores. Regarding the fine wines, W2 and W5, consumers perceived their smell as intense and complex, but the “initial impression” was lower than that of W1 and W4, eliciting low “expectation for the mouth” and “global evaluation” scores. The higher “thermal perception”, “body”, and “persistency” scores of W2 and W5 were not reflected in a higher “overall taste evaluation” score. Accordingly, the recognition of “evolution of the wine in the glass” and “duration of the wine fragrance”, with scores similar to those of W1 and W4, was not accompanied by a higher “global evaluation” score.

The appreciation of W3 elicited lower scores for almost all attributes, with unexpectedly high “smell complexity” and “impression in relation to odor” scores. Perhaps, the influence of perceived carbonation and residual sugar explains why these perceptions differed from what would be expected from a simple popular wine.

The PCA revealed further insight into the differences among the wines (Figure 2). W1 and W4 were placed in opposite quadrants from W2 and W5, evidencing the difference in the two wine styles. W3 was placed far from these wines due to negative scores in PC2.

Consumers recognized that W2 and W5 were more complex and persistent but did not rate these attributes as characteristic of high-quality wines. On the contrary, their preferences were directed to the fruity flavored wines, W1 and W4, whose initial attractiveness was not counteracted by mouthfeel perceptions. These results suggest that the low appeal of the yellow straw color and smell distaste are factors that prevent the recognition of positive scores of complexity and persistence as indicators of quality. Moreover, consumers recognized the higher persistence of W4 when compared with W1, consistent with their regional climate with Atlantic influence, but this was not reflected in a lower “global evaluation” score for W1. 

It is interesting to note that W2 and W5 both had negative scores in PC1, corresponding to lower scores with respect to the emotional attributes. However, consumers recognized their sensory features related to high-quality wines by placing them in the left quadrants of the plot. W1 and W4 were placed in the opposite side of the plot, while W3 was clearly separated from the other wines. 

The coefficients of correlation between the variables illustrate these observations (Appendix A). “Global evaluation” was positively correlated with “initial impression”, “expectation for the mouth”, and “overall taste evaluation”. On the contrary, it was negatively correlated with “smell complexity”. These results are consistent with the appeal of the intense fruity/flowery wines, W1 and W4, and their lower complexity when compared with W2 and W5. Furthermore, “global evaluation” was not related to “thermal perception”, “body”, astringency”, “persistency”, “evolution of the wine in the glass”, and “duration of wine fragrance”, meaning that consumers did not recognize these attributes as meriting higher likeability scores. Distaste for color and smell probably blurs the perception of quality attributes, such as “persistence”, “evolution of the wine in the glass”, and “duration of wine fragrance”. Given that these attributes were clearly recognized in W2 and W5, the question is how to make consumers understand that they are indicators of high-quality wines. If this goal is achieved, then wine professionals have the opportunity to expand the range of quality recognition by consumers.

A comparison between the perceptions of the semi-trained panel and the consumers may be performed by using the respective coefficients of determination (Appendix A). High correlations were obtained with “smell complexity”, “thermal perception”, and “duration of wine fragrance”, revealing agreement for each of these attributes between the two panels. These results evidence the attributes that do not require extensive training. Apparently, easier-to-define perceptions such as “smell intensity” and “persistency” were not correlated, evidencing the need for adequate training. Consumers’ “global evaluation” scores could not be predicted from any of the attributes evaluated by the semi-trained panel.

### 3.3. Check-All-That-Apply for Emotional Responses

The results of Cochran’s Q test for the emotional responses elicited by the five wines are shown in Table 5. A total of 15 emotional responses provided significant differences among the wines, either with positive or negative connotations. Interestingly, the first two emotions were of opposite nature (pleasant and aggressive), followed by four positive ones (joyful, warm, light, and chewable).

The most pleasant and joyful wines were W1 and W4 in contrast with W2, W3, and W5, consistent with their higher “global evaluation” scores. The most aggressive and least relaxed wines were W2 and W5, followed by W3, which is in accordance with their sensory features. Moreover, W2 and W5 were the warmest, related to their higher mouth thermal impression. These two wines were also the lightest, but this was not in accordance with their higher body perception, showing that this concept is not easy to define. W2 was distinguished from W5 because of its higher unpleasantness and for being less attractive.

The cluster analysis for the CATA emotions grouped the wines similarly to the tasting sheet attributes (Figure 3), separating W1 and W4 from W2 and W5.

Figure 4 shows that the emotional responses contributing to the “global evaluation” scores were mostly of positive reactions. Greedy is not easy to categorize as positive or negative, but high scores were given to W1 and low scores to W5, suggesting that consumers tended to associate this emotion with higher likeability, and thus, it should be considered a positive emotion in this case.

### 3.4. Ideal Wine Based on Elicited Emotions

Figure 5 shows the distribution of the wines in the PCA plot of the emotional attributes. The wines closer to the ideal wine were W1 and W4, as expected from the higher scores of positive emotions. On the contrary, the aged complex wines were in the area of predominantly negative emotions.

### 3.5. Emotional Wheel Presentation

The result of the organization of the emotional responses in a wheel format is shown in Figure 6. In this figure, the emotional lexicon is visualized as a hierarchical, segmented wheel, which presents the emotional attributes as basic emotions segmented according to their subordinated emotional grouping of emotional attributes. Outer-tier terms are the specific oral qualities that were used to score and describe white wine. 

Concerning negative affect emotions, the category “sickness” included the attributes “cloying” and “sickening”, based on its meanings and considering that these attributes referred explicitly to physical discomfort. Besides that, the positive correlation between these two attributes was weak, which indicated that they could not be considered synonyms. The attribute “unpleasant” presented a strong negative correlation to almost positive affect attributes, reinforcing their positive classification. Considering Laros and Steemkamp [26]’s classification structure, the attributes “unpleasant”, “melancholic”, and disappointing should be subordinated to “sadness”. However, the relevance of the negative correlation of unpleasant and the positive affect attributes justified the replacement of “sadness” with “unpleasantness”.

Concerning positive affect emotions, the attributes “sensual” and “passionate” were subordinated to the basic emotion “love”, though Laros and Steemkamp [26] did not include this emotion in their classification, implying that love is not an emotion related to consumption. However, wine consumption is permeated by several symbolic and cultural meanings [29], which expand emotional manifestation related to its use. Moreover, we contend that wine consumption involves specific symbolic connotations related to religion in Western culture, which are reflected in historical and social wine relevance. Ciolfi [30] observed that in wine consumption, emotional impressions are memorized and can be reactivated in future consumption experiences, even when remembered or imagined. Therefore, love can be accepted as an emotion in wine consumption.

## 4. Conclusions

The results presented in this work showed that an emotion-based tasting sheet enabled us to characterize different styles of white wines using a large consumer cohort. Indeed, consumers without previous training rated some sensory descriptors related to fine wines, such as “complexity” and “duration of wine fragrance”, similar to the semi-trained panel. However, this recognition was not reflected in high hedonic ratings, with wines with more intense fruity/flowery flavors and smooth mouthfeel typical of the international commercial style being preferred. Moreover, the distinction of wine styles using both sensory and emotional responses was more consistent than that obtained with the semi-trained panel. These results were in accordance with the findings of Coste et al. [19] concerning red wines, where a Burgundy Pinot Noir was differentiated from a great gold-awarded red wine by the consumers, while the semi-trained panel did not manage to do so with sensory descriptors. The differences among the responses of whites and reds rested upon the higher correlations between sensory and emotional responses and the higher number of significant emotions as measured by Cochran’s Q test in the case of white wines. These results indicate that tasters had more difficulty in establishing the differences among red wines, probably due to the effect of astringency, which dominates over other sensory features. 

Therefore, the issue of quality recognition was of a cognitive nature and not related to sensitivity or learning difficulties. Hopefully, the challenging task of expanding the quality range acknowledgeable by consumers may be simplified by the utilization of emotional responses related to wine sensory attributes as described in this work. Then, repeated exposure to unattractive wines from cool climate regions may increase familiarity and strengthen consumer fidelity. If this recognition is not achieved, the likely outcome is that, through the current technological options, winemakers will prefer to turn wines from cooler regions into products similar to their warm climate counterparts to guarantee commercial success.

## Figures and Tables

**Figure 1 foods-09-00452-f001:**
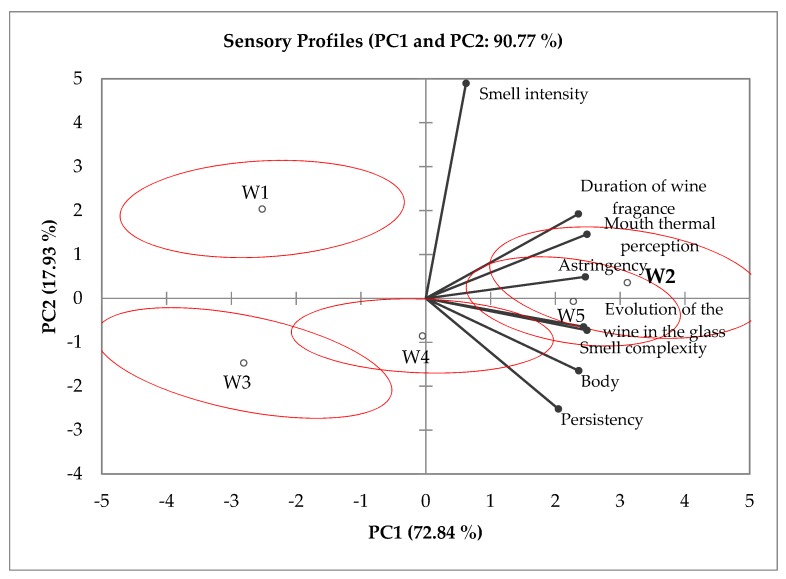
Two-dimensional map made from Principal Component Analysis (PCA) of the sensorial profiles of the white wines (W) obtained by ODP.

**Figure 2 foods-09-00452-f002:**
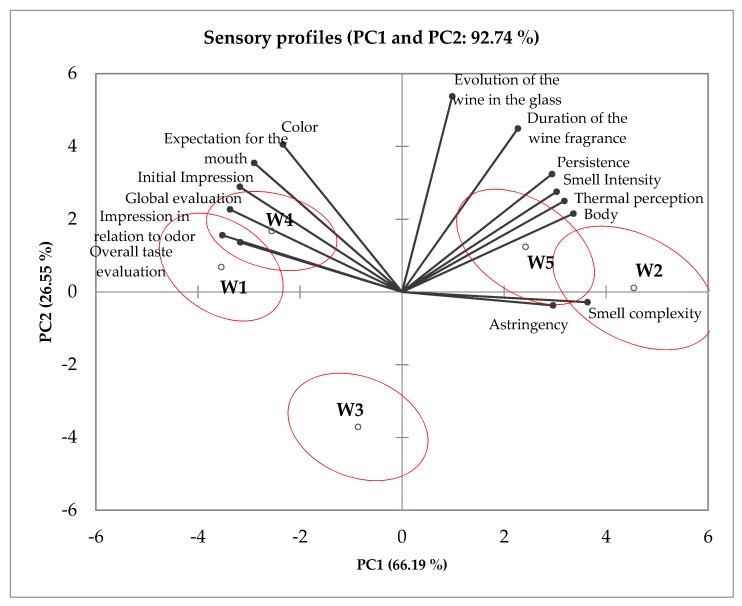
Two-dimensional map made from the PCA of the wine (W) descriptor scores using the emotional tasting sheet of the consumer panel.

**Figure 3 foods-09-00452-f003:**
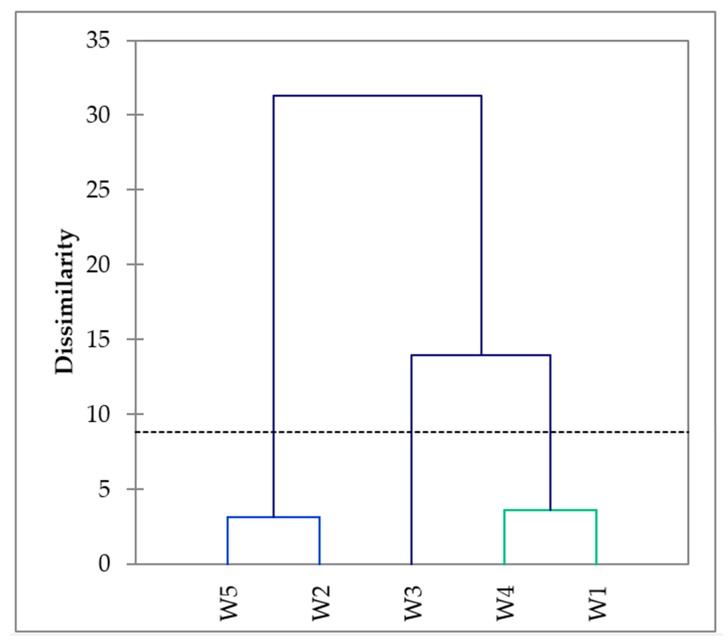
Dendrogram obtained from the Check-All-That-Apply (CATA) evaluation by the consumer panel of the white wines.

**Figure 4 foods-09-00452-f004:**
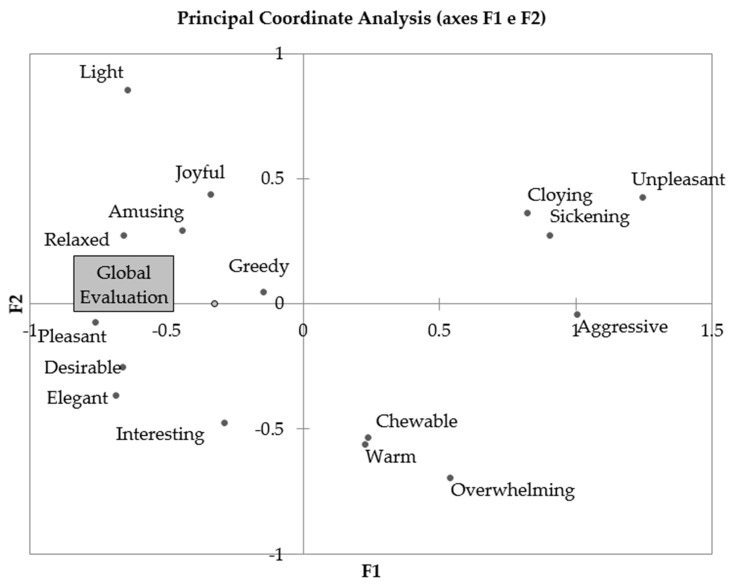
Principal Component Analysis of the results from the CATA evaluation combined with the “global evaluation” scores.

**Figure 5 foods-09-00452-f005:**
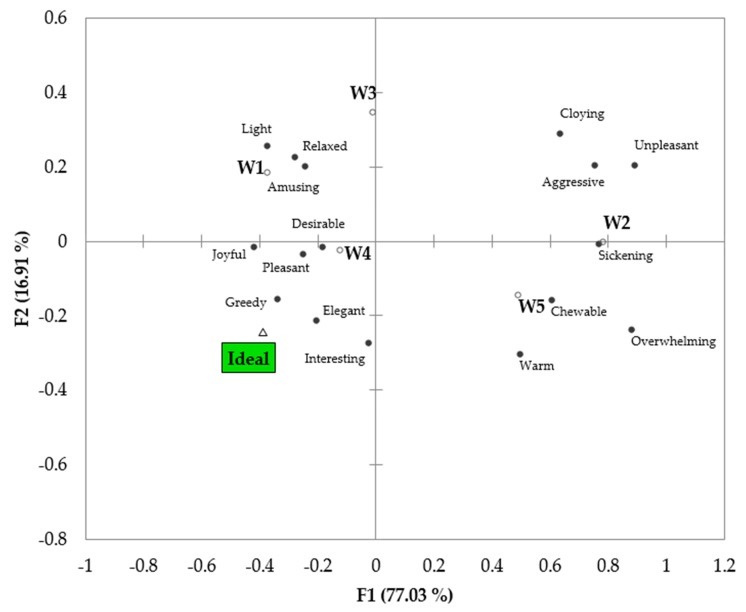
PCA plot of wines in relation to the ideal wine (W) according to emotional responses.

**Figure 6 foods-09-00452-f006:**
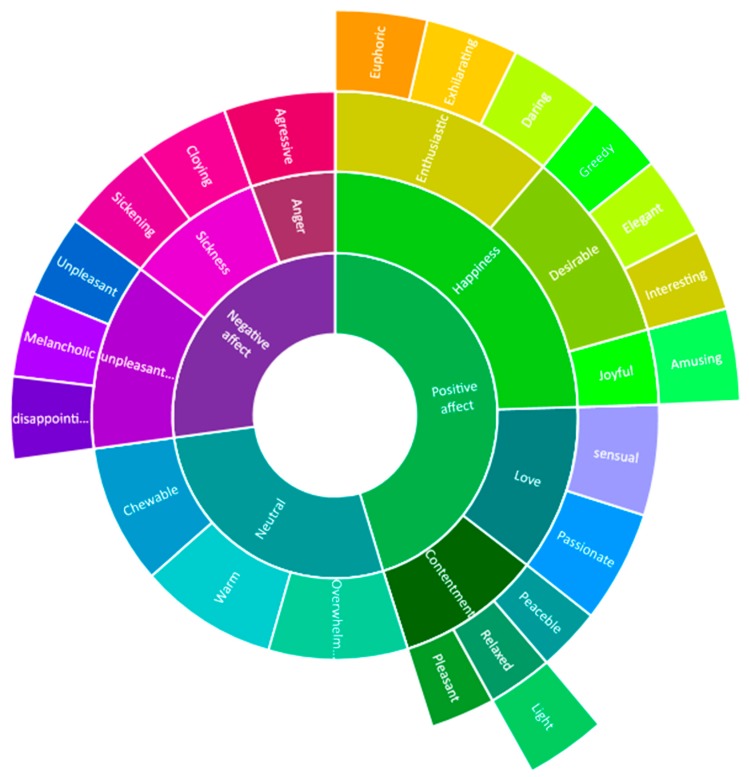
The wine taster’s emotion wheel.

**Table 1 foods-09-00452-t001:** Summary description of the Portuguese white wines used in the Optimized Descriptive Profile (ODP), Check-All-That-Apply (CATA) test, and in the consumer tasting.

Wine	W1	W2	W3	W4	W5
Brand/Producer	Vale dos Barris/Adega Cooperativa de Palmela	Colheita do Sócio Reserva/Adega Cooperativa da Covilhã	Casal Garcia/Aveleda	Casa da Senra/Estate producer	Pouilly-Fuissé/Roux Pére & Fils
Region	Península de Setúbal IPR ^a^	Beira Interior DOC ^b^	Vinho Verde DOC	Vinho Verde DOC	Pouilly-Fuissé (Burgundy) AOC ^c^
Varietal	Muscat of Alexandria	Blend	Blend	Loureiro	Chardonnay
Vintage	2014	2005	Not dated	2014	2004
Color	Citrus yellow	Yellow straw	Citrus yellow	Citrus yellow	Yellow straw
Odor	Intense flowery–fruity	Medium intensity developed	Medium intensity, fruity	Medium intensity, fruity	Medium intensity developed
Acidity (taste)	Low	Medium to high	Medium	Medium to high	High
Residual Sugars	<2 g/L	<2 g/L	11 g/L	<2 g/L	<2 g/L
Alcohol	12.0% vol.	13.5% vol.	9.5% vol.	13.0% vol.	13.0% vol.

^a^ IPR: Portuguese acronym for Protected Geographical Indication. ^b^ DOC: Portuguese acronym for Protected Denomination of Origin. ^c^ AOC: French acronym for Protected Appellation of Origin.

**Table 2 foods-09-00452-t002:** Values of F_sample × taster_ and significance levels for the sensory attributes of the white wines.

Attribute	Fsample × Taster	*p*-Value
Intensity (smell)	4.485	0.000 *
Thermal Perception	5.790	0.000 *
Body	3.621	0.000 *
Astringency	4.893	0.000 *
Persistence	3.590	0.000 *
Evolution of the Wine in the Glass	4.649	0.000 *
Duration of the Wine Fragrance	5.151	0.000 *
Complexity	7.611	0.000 *

* Significant at 5% probability.

**Table 3 foods-09-00452-t003:** Mean scores for each attribute in the ODP.

Attributes	*p*-Values	Wines				
		W1	W2	W3	W4	W5
Smell Intensity	<0.0001	5.184 ^c^	4.086 ^bc^	2.269 ^a^	3.631 ^ab^	4.247 ^bc^
Smell Complexity	<0.0001	1.876 ^a^	5.678 ^d^	2.647 ^ab^	3.233 ^bc^	4.400 ^cd^
Mouth Thermal Perception	<0.0001	2.824 ^a^	4.669 ^b^	2.111 ^a^	2.862 ^a^	4.238 ^b^
Body	0.001	2.189 ^a^	3.516 ^bc^	2.482 ^ab^	3.482 ^bc^	3.758 ^c^
Astringency	0.001	1.949 ^ab^	2.991 ^bc^	1.700 ^a^	2.284 ^abc^	3.396 ^c^
Mouth Persistence	0.002	2.544 ^a^	3.845 ^ab^	3.122 ^ab^	4.211 ^b^	4.316 ^b^
Evolution of the Wine in the Glass	0.000	1.657 ^a^	3.881 ^b^	2.082 ^a^	2.416 ^a^	2.941 ^ab^
Duration of the Wine Fragrance	<0.0001	3.441 ^ab^	5.419 ^c^	2.250 ^a^	3.630 ^ab^	4.191 ^bc^

Notes: Minimum, score 0; maximum, score 9; numbers in the same row followed by the same letter, or a pair of letters, are not statistically different (*p* < 0.05).

**Table 4 foods-09-00452-t004:** Estimated mean values of each attribute of the tasting sheet for wines W1 to W5 by the consumer panel.

Descriptors	*p*-Values	Wines				
		W1	W2	W3	W4	W5
Color	<0.0001	3.625 ^bc^	3.202 ^a^	3.212 ^a^	3.942 ^c^	3.500 ^ab^
Initial Impression	<0.0001	3.654 ^b^	2.689 ^a^	2.903 ^a^	3.692 ^b^	3.019 ^a^
Smell Intensity	<0.0001	3.279 ^a^	3.808 ^b^	2.971 ^a^	3.135 ^a^	3.702 ^b^
Smell Complexity	0.003	2.845 ^a^	3.519 ^b^	3.154 ^ab^	2.990 ^ab^	3.417 ^b^
Expectation for the Mouth	0.003	3.375 ^bc^	2.902 ^a^	2.923 ^ab^	3.394 ^c^	3.058 ^abc^
Impression in Relation to Odor	0.004	3.275 ^b^	2.827 ^a^	3.165 ^ab^	3.388 ^b^	3.176 ^ab^
Thermal Perception	<0.0001	2.288 ^a^	3.317 ^c^	2.385 ^ab^	2.750 ^b^	3.437 ^c^
Body	<0.0001	2.269 ^a^	3.346 ^b^	2.375 ^a^	2.631 ^a^	3.346 ^b^
Astringency	<0.0001	2.269 ^a^	3.067 ^b^	2.837 ^b^	2.837 ^b^	2.990 ^b^
Persistence	<0.0001	3.019 ^a^	3.702 ^b^	2.913 ^a^	3.327 ^ab^	3.500 ^b^
Overall Taste Evaluation	<0.0001	3.558 ^b^	2.865 ^a^	3.212 ^ab^	3.548 ^b^	3.144 ^ab^
Evolution of the Wine in the Glass	0.002	2.806 ^b^	2.846 ^b^	2.327 ^a^	2.769 ^b^	2.856 ^b^
Duration of the Wine Fragrance	0.000	3.223 ^ab^	3.519 ^b^	2.875 ^a^	3.269 ^ab^	3.500 ^b^
Global Evaluation	<0.0001	3.853 ^b^	3.030 ^a^	3.279 ^a^	3.825 ^b^	3.252 ^a^

Note: numbers in the same row followed by the same letter, or a pair of letters, are not statistically different (*p* ≤ 0.05).

**Table 5 foods-09-00452-t005:** Cochran’s Q test for each emotion of the CATA list.

Emotions	*p*-Values	Wines				
		W1	W2	W3	W4	W5
Pleasant	0.000	0.548 ^b^	0.202 ^a^	0.308 ^a^	0.452 ^b^	0.260 ^a^
Aggressive	0.000	0.077 ^a^	0.356 ^d^	0.231 ^bc^	0.154 ^ab^	0.288 ^cd^
Joyful	0.000	0.221 ^b^	0.029 ^a^	0.202 ^b^	0.250 ^b^	0.077 ^a^
Warm	0.000	0.096 ^ab^	0.337 ^c^	0.058 ^a^	0.183 ^b^	0.337 ^c^
Light	0.000	0.510 ^c^	0.096 ^a^	0.365 ^b^	0.308 ^b^	0.087 ^a^
Chewable	0.001	0.019 ^a^	0.173 ^c^	0.058 ^ab^	0.067 ^ab^	0.106 ^bc^
Unpleasant	0.000	0.048 ^a^	0.250 ^b^	0.106 ^a^	0.058 ^a^	0.125 ^a^
Relaxed	0.000	0.365 ^d^	0.087 ^a^	0.212 ^bc^	0.250 ^cd^	0.106 ^ab^
Desirable	0.002	0.317 ^b^	0.135 ^a^	0.231 ^ab^	0.327 ^b^	0.202 ^ab^
Interesting	0.005	0.212 ^abc^	0.183 ^ab^	0.125 ^a^	0.298 ^bc^	0.308 ^c^
Sickening	0.040	0.019 ^a^	0.077 ^ab^	0.038 ^ab^	0.019 ^a^	0.087 ^b^
Overwhelming	0.016	0 ^a^	0.067 ^b^	0.010 ^ab^	0.048 ^ab^	0.067 ^b^
Amusing	0.017	0.163 ^b^	0.029 ^a^	0.144 ^b^	0.125 ^b^	0.087 ^ab^
Cloying	0.033	0.096 ^a^	0.240 ^b^	0.173 ^ab^	0.115 ^a^	0.183 ^ab^
Greedy	0.049	0.135 ^b^	0.058 ^ab^	0.058 ^ab^	0.077 ^ab^	0.038 ^a^
Elegant	0.058	0.231 ^a^	0.144 ^a^	0.144 ^a^	0.269 ^a^	0.163 ^a^
Peaceable	0.165	0.202 ^b^	0.087 ^a^	0.135 ^ab^	0.154 ^ab^	0.115 ^ab^
Daring	0.204	0.106 ^a^	0.192 ^a^	0.202 ^a^	0.212 ^a^	0.173 ^a^
Disappointing	0.226	0.106 ^a^	0.183 ^a^	0.125 ^a^	0.096 ^a^	0.173 ^a^
Melancholic	0.245	0.067 ^ab^	0.106 ^b^	0.077 ^ab^	0.029 ^a^	0.077 ^ab^
Euphoric	0.310	0.048 ^a^	0.077 ^a^	0.115 ^a^	0.067 ^a^	0.058 ^a^
Sensual	0.406	0.135 ^a^	0.087 ^a^	0.058 ^a^	0.106 ^a^	0.106 ^a^
Exhilarating	0.713	0.125 ^a^	0.106 ^a^	0.087 ^a^	0.144 ^a^	0.125 ^a^
Passionate	0.773	0.096 ^a^	0.067 ^a^	0.067 ^a^	0.096 ^a^	0.106 ^a^
Surprising	0.860	0.106 ^a^	0.154 ^a^	0.135 ^a^	0.144 ^a^	0.125 ^a^

Note: numbers in the same row followed by the same letter, or a pair of letters, are not statistically different (*p* < 0.05).

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
