# Peer review of "Consumers Associate High-Quality (Fine) Wines with Complexity, Persistence, and Unpleasant Emotional Responses"

_foods, 2020, doi:10.3390/foods9040452_

Round 1

Reviewer 1 Report

This paper applies to a set of white wines a methodology, for wine evaluation through sensory and emotional consumers responses, previously develloped by the authors and applied to red wines. New and innovative instruments of analysis are also used in this case (wine taster's emotion wheel).

The approach is original and seems to be promising. It is an interesting contribution to understand a complex problem, and thus the content is significant.

Mostly it is well written and clear. However a revision is required to correct some minor misspelling or some unclear sentences. As for example:

Page 2, 3rd paragraph - Punctuation should be revised in order to make it clear.

In the same paragraph, 3rd line, it should be "it is understandable".

Page 4, last paragraph - The complete reference for Coste et al, is missing.

Page 12, Table 7 - Although  the notes below the table refer values in bold, there is none in the table.

Page 16, Figure 4 - one of the words in the title of the Figure is not an english word, I think it should be axes. 

Pages 18 and 19 - The description of the wine taster's emotion wheel development and discussion would benefit from some revision to make it easier to follow.

The paragraph below the wheel should be revised to make it understandable.

References - There are minor inconsistences, for example punctuation in reference 11 or 36  or the use of bold in reference 34.

Author Response

Response to Reviewer 1 Comments

All corrections have been made.

Text has been revised English.

Page 2, 3rd paragraph - Punctuation should be revised in order to make it clear.

- Changes were made in the punctuation throughout the text.

In the same paragraph, 3rd line, it should be "it is understandable".

- Change was made to "it is understandable".

Page 4, last paragraph - The complete reference for Coste et al, is missing.

The reference Coste et al. was completed.

Page 12, Table 7 - Although  the notes below the table refer values in bold, there is none in the table.

Table 7 was corrected.

Page 16, Figure 4 - one of the words in the title of the Figure is not an English word, I think it should be axes. 

- The change was made to "axes" in Figure 4.

Pages 18 and 19 - The description of the wine taster's emotion wheel development and discussion would benefit from some revision to make it easier to follow.

The description of the wine taster's emotion wheel development and discussion has been revised.

The paragraph below the wheel should be revised to make it understandable.

The paragraph has been revised.

References - There are minor inconsistencies, for example, punctuation in reference 11 or 36  or the use of bold in reference 34.

- Corrected accordingly.

Reviewer 2 Report

Referee's Comments:

The Article “Consumers associate high quality (fine) wines with complexity, persistence and unpleasant emotional responses” according to authors provides information of the application of a wine tasting sheet including sensory and emotional responses to simplify the recognition of fine white wines by consumers.

This is an interesting study with application in winemaking.

Suggestions:

In this regard, the following are suggestions to keep in mind to improve the quality of the manuscript text.

Abstract:

Authors should improve structure abstract and include an introduction. Please they rewrite the main objective of the study and review the numerical values provides. In the material and methods section they indicate a group of 103 consumers and the abstract the group was 104.

The letter ‘p’ of p-values should be always written in italics.

Introduction:

With regard to the aims of the study, it would be necessary to rewrite them to make them clearer and more accurate.

Materials and Methods:

In Focus Group Sessions section, include the emotional tasting sheet and CATA list obtained as a result of the focus group discussions. Have both normality and homogeneity been checked before applying two-way ANOVA? Please, indicate the tests used for it.

Results and Discussion

The letter ‘p’ of p-values should be always written in italics.

The authors included many tables, they could incorporate some of the tables as supplementary.

The wine laster's emotion wheel development should be moved to the Materials and Methods section.

Author Response

Response to Reviewer 2 Comments

All corrections have been made.

The text has been revised English.

1. Abstract:

Authors should improve structure abstract and include an introduction. Please they rewrite the main objective of the study and review the numerical values provides.

- An improvement in the explanation was introduced in the abstract, including an introduction and changing the conclusion.

In the material and methods section, they indicate a group of 103 consumers and the abstract the group was 104.

- Corrected accordingly to 104 in the material and methods section.

The letter ‘p’ of p-values should be always written in italics.

- Corrected accordingly.

2. Introduction:

With regard to the aims of the study, it would be necessary to rewrite them to make them clearer and more accurate.

- The aim of the study was rewritten.

3. Materials and Methods:

In Focus Group Sessions section, include the emotional tasting sheet and CATA list obtained as a result of the focus group discussions.

- The emotional tasting sheet and CATA list obtained as a result of the focus group discussions were included such as Figure 1S in Supplementary Material.

Have both normality and homogeneity been checked before applying two-way ANOVA? Please, indicate the tests used for it.

- The tests used were added in Material and Methods.

4. Results and Discussion

The letter ‘p’ of p-values should be always written in italics.

- Corrected accordingly.

The authors included many tables, they could incorporate some of the tables as supplementary.

Tables 4, 6 and 7 were incorporated as supplementary.

The wine laster's emotion wheel development should be moved to the Materials and Methods section.

The wine laster's emotion wheel development was moved to the Materials and Methods section.